The level of putative carotenoid-binding proteins determines the body color in two species of endemic Lake Baikal amphipods

Drozdova Polina drozdovapb@gmail.com 1 2
Saranchina Alexandra 1
Morgunova Mariya 1
Kizenko Alena 3 4
Lubyaga Yulia 1 2
Baduev Boris 1
Timofeyev Maxim m.a.timofeyev@gmail.com 1 2
1 Institute of Biology, Irkutsk State University , Irkutsk , Russia
2 Baikal Research Centre , Irkutsk , Russia
3 Institute of Cytology RAS , St. Petersburg , Russia
4 Bioinformatics Institute , St. Petersburg , Russia
Ford Alex
Electronic publication date: 2020 Jun 19
Publication date: 2020
Volume: 8
Electronic Location ID: e9387
Received 2020 Jan 15; Accepted 2020 May 28
Copyright: ©2020 Drozdova et al.
Copyright year: 2020
Copyright holder: Drozdova et al.
License: This is an open access article distributed under the terms of the Creative Commons Attribution License, which permits unrestricted use, distribution, reproduction and adaptation in any medium and for any purpose provided that it is properly attributed. For attribution, the original author(s), title, publication source (PeerJ) and either DOI or URL of the article must be cited.
License URL: https://creativecommons.org/licenses/by/4.0/

Keywords: Amphipoda, Baikal, Amphipod coloration, Crustacyanins, Carotenoid-binding proteins

Funding: Russian Science Foundation 19-74-00045 Ministry of Science and Higher Education of the Russian Federation FZZE-2020-0026 This study was supported by the Russian Science Foundation (grant number 19-74-00045) that funded the sampling, laboratory analyses, and the work of Polina Drozdova and Alexandra Saranchina. The work of Mariya Morgunova, Yulia Lubyaga, Boris Baduev, and Maxim Timofeyev was supported by the Ministry of Science and Higher Education of the Russian Federation (project N FZZE-2020-0026). There was no additional external funding received for this study. The funders had no role in study design, data collection and analysis, decision to publish, or preparation of the manuscript.

==============================
Color is an essential clue for intra- and interspecies communication, playing a role in selection and speciation. Coloration can be based on nanostructures and pigments; carotenoids and carotenoproteins are among the most widespread pigments in animals. Over 350 species and subspecies of amphipods (Crustacea: Amphipoda) endemic to Lake Baikal exhibit an impressive variability of colors and coloration patterns, including intraspecific color morphs. However, the mechanisms forming this diversity are underexplored, as while the carotenoid composition of several transparent, green, and red species was investigated, there have been no reports on the corresponding carotenoid-binding proteins. In this work, we analyze the coloration of two brightly colored Baikal amphipods characterized by intraspecific color variability, Eulimnogammarus cyaneus and E. vittatus. We showed that the color of either species is defined by the level of putative carotenoid-binding proteins similar to the pheromone/odorant-binding protein family, as the concentration of these putative crustacyanin analogs was higher in blue or teal-colored animals than in the orange- or yellow-colored ones. At the same time, the color did not depend on the total carotenoid content, as it was similar between animals of contrasting color morphs. By exploring the diversity of these sequences within a larger phylogeny of invertebrate crustacyanins, we show that amphipods lack orthologs of the well-studied crustacyanins A and C, even though they possess some crustacyanin-like sequences. The analysis of expression levels in E. cyaneus showed that the transcripts encoding crustacyanin analogs had much higher expression than the crustacyanin-like sequences, suggesting that the former indeed contribute to the color of these brightly colored animals. The crustacyanin analogs seem to act in a similar way to the well-studied crustacyanins in body color formation, but the details of their action are still to be revealed.

Introduction

Color is an important visual clue for many groups of organisms. The observed variety of colors is based on two principles, which are structural coloration and pigments (Cuthill et al., 2017). Carotenoids are one of the most widespread pigments in animal coloration, even though the absolute majority of animal species cannot synthesize carotenoids de novo and thus rely on their presence in the food source (Maoka, 2020).

Lake Baikal is home to over 350 endemic species and subspecies of gammaridean amphipods (Crustacea: Malacostraca: Amphipoda), which constitutes about half of the diversity of the amphipod fauna of surface freshwaters (Takhteev, 2019). These species differ significantly by preferred habitats and appearance. Different species are adapted to diverse conditions from the water edge to the depths below 1,500 m (Takhteev, Berezina & Sidorov, 2015), but their morphological diversity cannot be explained solely by adaption to habitat. One of the most evident aspects of this diversity is the pronounced variability of body color and coloration pattern (Fig. 1). The overall intensity of body color varies from almost transparent (e.g., Macrohectopus branickii (Dybowsky, 1874)) or white species (e.g., Ommatogammarus albinus (Dybowsky, 1874)) to blood red (Eulimnogammarus cruentus ((Dorogostaysky, 1930)), dark olive-green (e.g., E. verrucosus (Gerstfeldt, 1858) or Pallasea cancellus (Pallas, 1776)), blue or orange (E. cyaneus (Dybowsky, 1874) or violet-blue (E. czerskii (Dybowsky, 1874). For some species (including mass littoral species E. vittatus (Dybowsky, 1874), E. cyaneus, and E. messerschmidtii Bedulina et Takhteev 2014), subspecific color morphs are known (Timoshkin, 2001; Bedulina et al., 2014). However, the molecular mechanisms underlying color formation in Baikal endemic amphipods are underexplored. Only the carotenoid composition of several transparent, green, and red species was investigated (Czeczuga, 1975; Dembitsky & Rezanka, 1996), but these studies did not find a mechanism allowing for the formation of this coloration diversity.

Figure 1 Representative photographs of some Lake Baikal endemic amphipod species.

(A) M. branickii (Dybowsky, 1874); (B) Brandtia latissima latior (Dybowsky, 1874); (C) Micruropus wohlii wohlii (Dybowsky, 1874); (D) Gmelinoides fasciatus (Stebbing, 1899); (E) E. verrucosus (Gerstfeldt, 1858); (F–G) different color morphs of E. cyaneus (Dybowsky, 1874); (H–J) different color morphs of E. vittatus (Dybowsky, 1874); (K) E. maackii (Gerstfeldt, 1858); (L–M) different color morphs of E. messerschmidtii (Bedulina et Takhteev, 2014); (N) E. ussolzewii ussolzewii (Dybowsky, 1874); (O) E. cruentus (Dorogostaisky, 1930); (P) O. albinus (Dybowsky, 1874); (Q) O. flavus (Dybowsky, 1874); (R) O. carneolus melanophthalmus (Bazikalova, 1945). Photo credit (E, O): Kseniya Vereshchagina.

The major player determining morphological body coloration in crustaceans are carotenoid pigments, especially astaxanthin and its derivatives. The level of carotenoids was shown to be correlated with the body color in several amphipod species of the genus Gammarus (Hindsbo, 1972; Gaillard et al., 2004), in which carotenoid depletion due to acanthocephalan infection leads to a change in color. However, the relationship between the total carotenoid content and body color is not as simple. In shrimps, carotenoid content may also be the reason behind the characteristic color of some morphs, as albino Fenneropenaeus merguiensis individuals had the lowest astaxanthin content (Ertl et al., 2013), but in other cases, the body color did not correlate with the total carotenoid content (Ertl et al., 2013; Tume et al., 2009).

Among the factors adding complexity to color determination are the distribution of carotenoids in the epithelial tissue (Wade et al., 2015), carotenoid composition and carotenoid-binding proteins. These proteins expand the palette of carotenoid-based colorations to cover the whole spectrum from red and orange to blue and purple (Maoka, 2011). The best-studied example of such proteins is the lobster (Homarus gammarus) shell protein, crustacyanin, which binds to astaxanthin and provides the lobster carapace with its characteristic blue color (Buchwald & Jencks, 1968; Chayen et al., 2003). Crustacyanins, belonging to the lipocalin family and found in decapods and stomatopods, appear to be a strictly crustacean-specific innovation (Wade et al., 2009). Apart from lobster species, they were explored on the sequence level in other decapods, mainly penaeid shrimps (Ertl et al., 2013; Budd et al., 2017). Decapod crustacyanins form two groups, A and C, which (at least in lobster) form heterodimers called β-crustacyanin binding to two astaxanthin molecules each, and eight β-crustacyanin subunits form one α-crustacyanin molecule (Chayen et al., 2003).

However, the structural and functional diversity of crustacyanins in amphipods remains almost unknown: some crustacyanin-like sequences, forming a sister group to both A and C groups of decapods, were found in expressed sequence tags of Gammarus pulex, but their identity as crustacyanins remained unclear (Wade et al., 2009). In addition, two crustacyanin-like proteins were isolated from G. lacustris with ion-exchange chromatography (Czeczuga & Krywuta, 1981), but the authors identified only the amino acid composition and not the sequences of these proteins. So, while it is logical to suggest the existence of proteins acting as crustacyanins in amphipods, their sequences and thus evolutionary origin remain unknown.

The goal of this work was to uncover the mechanism underlying the color formation in two endemic Baikal amphipod species, E. cyaneus and E. vittatus. We estimated the carotenoid content in individuals of different color morphs, characterized the putative carotenoid-binding proteins analogous to crustacyanins, and placed them in the larger phylogeny of the invertebrate coloration-related proteins.

Materials and Methods

Animals and sampling

Eulimnogammarus cyaneus (Dybowsky, 1874) is a relatively small (adult body size 11–15 mm) species widespread around the shoreline of Lake Baikal. It occupies the depths from the water edge to several meters, concentrating near the shoreline (Bazikalova, 1945). The original description of the species (as G. cyaneus) described its color as greyish blue (schmutzig blau) (Dybowsky, 1874). The same epithet was reproduced when the species was reassigned to the genus Eulimnogammarus (Bazikalova, 1945). A newer source describes its color as “continuous variation from sky blue to bluish-green, then with orange-red antennae, and to fully orange individuals.” This index also notes that the ratio of different color morphs varies along the coast, but precopulae exist in all possible variations, and the prevailing morph was bluish-green with orange antennae (Timoshkin, 2001). Our observations agree with these facts (Fig. S1). Moreover, allozyme analysis showed that the orange and blue individuals appear to form panmictic populations in all studied locations (Mashiko et al., 2000), again confirming the intraspecies nature of this color polymorphism.

E. vittatus (Dyboswky, 1874) is a slightly bigger (adult body size 18–20 mm) species also widespread in Lake Baikal littoral and found at up to 30-m depths, but concentrating mostly at depth up to 2–3 m (Bazikalova, 1945). The original description of this species (as G. vittatus) defined its color as light yellowish-green or light olive green with brownish stripes in the hind part of each segment (Dybowsky, 1874). Newer sources (Timoshkin, 2001) note that the live color of E. vittatus varies greatly as different shades of yellow, blue, and green, with the dark stripes of the hind part of each segment being the common characteristic of all color morphs.

Most E. cyaneus individuals were sampled in August 2019 in Bolshie Koty (south-west coast of Baikal; 51°54′11.67″N 105°4′7.61″E). Some photographs feature animals sampled near Listvyanka (51°52′14.07″N 104°49′41.78″E) in July 2019. The individuals of E. vittatus were sampled in Listvyanka in April 2019 and January 2020. The animals were caught with kick sampling in Lake Baikal littoral at depths of 0–0.5 m and transported to the laboratory in insulated boxes. In the laboratory, they were kept in 2 l plastic tanks with Baikal water and several sterilized Baikal stones per tank at 8 ± 2 °C under constant aeration and fed ad libitum with a dried and ground mixture of invertebrates and macrophytes from their habitat. The water was exchanged once in three days. The blue and orange E. cyaneus individuals caught in August were sorted immediately after sampling. All (103) orange individuals and approximately the same number of blue ones (93 individuals) were kept in the same tank for three weeks to normalize the environmental conditions before taking photographs and fixation.

Photographs

All photographs used for color quantification were taken with an Olympus Tough TG-5 camera (Olympus, China) in the microscope mode against the same white background. One blue and one orange individual were included in each photograph to compensate for any unnoticed effects of poor color balance. The photograph was loaded into the GIMP software (https://gimp.org), and white balance was corrected against the white background with the Levels tool. Red, blue and green color values were recorded with the Color Picker Tool in GIMP from the pereon (approx. 6th segment), gut (the best visible segment), pereopods, and antennae (whichever was the most clearly visible). An example is shown in Fig. S2. The R/B ratio was subsequently used as a color index, similarly to how it was applied to study color morphs of the coconut crab (Nokelainen, Stevens & Caro, 2017).

Some photographs were also taken with the Altami SPM0880 stereo microscope (Altami, Russia) equipped with a camera (U3CMOS05100KPA, Altami, Russia); the white balance was auto-corrected in the Altami Studio software prior to shooting against a 17% grey paper.

Animal fixation

For most samples, the hemolymph was extracted with glass capillaries and immediately mixed with anti-coagulation buffer (Shchapova et al., 2019) (∼1:1-1.5 volume/volume), appendages were fixed in 96% ethanol, and the rest of the sample was shock frozen in liquid nitrogen.

Carotenoid measurements

Carotenoid concentration was assessed with a spectrophotometry-based method based on the published procedures (British Standards Institute, 1977; Johnston et al., 2000; Razi Parjikolaei et al., 2015) with modifications. Samples (either one whole E. vittatus individual or E. cyaneus individual devoid of hemolymph and several appendages, in both cases shock-frozen in liquid nitrogen and stored at −80 °C) were put in the water near to the boiling temperature for 7–10 s until the color changed to orange, dried with a paper towel and weighed. Then, carotenoids were extracted by homogenizing the samples in 1.5–3 ml acetone (Vekton, Russia) with stainless steel beads (Qiagen, Germany) using a Tissue lyser (Qiagen, Germany) in three consecutive rounds with default settings (50 rpm for 2 min). Each time, the debris was pelleted, and the supernatant was collected into glass tubes. Then, the collected supernatant was mixed with 0.5–1 ml petroleum ether 40-70 (Ekos-1, Russia), and at least 10 ml of distilled water was added to the mixture. After phase separation, the absorbance of the non-polar upper fraction was measured at the wavelengths from 200 to 800 nm with Cary 50 UV/VIS spectrophotometer (Varian Inc., Belrose, Australia). The purity of the extract was controlled by absorbance at 600 nm, and the concentration of carotenoids in parts per million (ppm) was estimated based on the absorbance at 450 nm (A450) as 4 × A450 × V / M, where V is the volume of petroleum ether used for re-extraction (ml) and M is the wet sample weight (g) (British Standards (Institute, 1977).

Protein extraction and electrophoretic methods

Hemolymph was used as the source for protein extraction, as it contained less distinct proteins than the whole body extract, but as the color of the hemolymph generally matched the body color (see the Results section). For one-dimensional polyacrylamide gel electrophoresis (1D-PAGE), we added an equal amount of 2×sampling buffer (Laemmli, 1970) to the hemolymph/anti-coagulation buffer mixture (hemolymph of one animal was used), incubated it at 95 °C for 2 min, chilled on ice and loaded into 12% acrylamide gel blocks. The gels were run at 60 V for approximately 30 min and then at 120 V until the dye reached the end of the gel, according to the standard procedure (Sambrook, Fritsch & Maniatis, 1989). The PageRuler Prestained Protein Ladder, 10 to 180 kDa (Thermo Scientific, USA) was used to assess protein molecular weights.

Protein purification from hemolymph and two-dimensional PAGE (2D-PAGE) was performed according to the published method (Bedulina et al., 2016), except for the fact that cells were not pelleted, as they contribute only a small fraction of protein to the hemolymph. The hemolymph of ten E. cyaneus or four E. vittatus individuals was pooled in each sample. Isoelectric focusing was run as described (Bedulina et al., 2016; Bedulina et al., 2017), and separation by molecular weight was also run according to this protocol, except for the fact that smaller gels were used in the case of E. cyaneus.

Native 2D-PAGE was run using essentially the same protocol, except for the fact that sodium dodecyl sulfate (SDS) and beta-mercaptoethanol were omitted from buffers, and the hemolymph was not heated before loading. After native electrophoresis, the bands of interest were cut out, incubated in loading buffer with SDS for at least 30 min, loaded into the wells of a regular polyacrylamide gel and run under denaturing conditions.

All gels were stained with 0.2% Coomassie Brilliant Blue in 10% acetic acid / 25% ethanol and destained with hot distilled water. Gel densitometry was performed with the ImageJ/Fiji package (Schindelin et al., 2012; Schneider, Rasb & Eliceiri, 2012). The relative abundances of two putative crustacyanin spots were calculated as the ratios between the integrated optical density of the corresponding spot to the sum of integrated optical density values for the two spots and the major hemocyanin spot. The bands of interest were cut from gels with a scalpel in sterile conditions for subsequent identification with liquid chromatography with tandem mass spectrometry (LC-MS/MS).

LC-MS/MS analysis

Proteins were subjected to in-gel trypsin digestion according to the following procedure. After three washes in water, the gel pieces were incubated in 50% (v/v) acetonitrile and 100 mM ammonium bicarbonate (pH 8.9) for 20 min, then in 100% acetonitrile for 20 min. The pieces were dried for 1 h. Depending on the original size of the gel slice, 5–8 µl of trypsin solution (25 ng/µl sequencing grade modified trypsin (Promega, Madison, WI, USA) in 50 mM ammonium bicarbonate solution) were added, and protein hydrolysis was carried out at 37 °C overnight. Tryptic peptides were extracted by the addition of 15 µl extraction solution (5% acetonitrile, 0.5% formic acid) for 30 min and analyzed by LC-MS/MS.

The peptides were separated with high-performance liquid chromatography (Ultimate 3000 Nano LC System, Thermo Scientific, Rockwell, IL, USA) in a 15-cm long C18 column with an inner diameter of 75 µm (Acclaim® PepMap™ RSLC, Thermo Fisher Scientific, Rockwell, IL, USA). The peptides were eluted with a gradient from 5–35% buffer B (80% acetonitrile, 0.1% formic acid) over 45 min at a flow rate of 0.3 µL/min. Total run time including 5 min to reach 99% buffer B, flushing 5 min with 99% buffer B and 5 min re-equilibration to buffer A (0.1% formic acid) was 60 min.

MS analysis was performed in triplicate with a Q Exactive HF mass spectrometer (Q Exactive™ HF Hybrid Quadrupole-Orbitrap™ Mass spectrometer, Thermo Fisher Scientific, Rockwell, IL, USA). Mass spectra were acquired at a resolution of 120,000 (MS) and 15,000 (MS/MS) in an m/z range of 350–1,500 (MS) and 100–2,000 (MS/MS). Isolation threshold of 100,000 counts was determined for precursor selection, and up to top 10 precursors were chosen for fragmentation with high-energy collisional dissociation (HCD) at 30 NCE and 100 ms accumulation time. Precursors with a charged state of +1 were rejected, and all measured precursors were excluded from measurement for 20 s. The mass spectrometry proteomics data have been deposited to the ProteomeXchange Consortium (Deutsch et al., 2020) via the PRIDE (Perez-Riverol et al., 2019) partner repository with the dataset identifier PXD018516 and 10.6019/PXD018516

Data analysis and bioinformatic methods

The comparison of relative protein abundances and carotenoid content was performed in the R statistical environment (R Core Team, 2019) v3.6.1 and visualized with the ggplot2 package (Wickham, 2016) v3.2.1 for R. Groups of samples were compared with the Mann–Whitney rank-sum test with Holm correction for multiple comparisons where applicable.

The blastx command from the ncbi-blast+ package (Camacho et al., 2009) v2.2.28+ was used to search for the sequences of H. gammarus crustacyanins A2 (P80007; Keen et al., 1991a) and C1 (P80029; Keen et al., 1991b) in the published transcriptome assemblies of the studied species (GEPS01 and GEPV01 from (Naumenko et al., 2017); GHHW01 from (Drozdova et al., 2019); NCBI IDs of the assemblies are shown here and elsewhere), as well as in the reassembled data. Trinity (Grabherr et al., 2011) v2.8.5 was used to reassemble the E. vittatus transcriptome from the published raw sequencing reads (Naumenko et al., 2017); SRA NCBI: SRR3467061). The expression level of the transcripts was estimated with salmon (Patro et al., 2017) v0.12.0 with the wrapper script from Trinity v2.8.5. Protein sequences encoded by transcripts were predicted with the getorf function of the emboss package (Rice, Longden & Bleasby, 2000) v6.6.0.0. The diamond package (Buchfink, Xie & Huson, 2014) v0.9.23.124 was used to re-classify the found sequences against the NCBI non-redundant protein database (Oct 10, 2017). The results were visualized in the R statistical environment (R Core Team, 2019) with the ggplot2 package (Wickham, 2016).

Protein sequence alignment was performed with prank (Löytynoja, 2014) v.170427. The alignments were trimmed with trimal (Capella-Gutierrez, Silla-Martinez & Gabaldon, 2009) v1.4.rev22 and analyzed with iqtree (Nguyen et al., 2014) v1.6.12 to reconstruct the phylogeny; model selection was performed automatically with ModelFinder (Kalyaanamoorthy et al., 2017), and the topology was tested using 1,000 Shimodaira-Hasegawa approximate likelihood ratio test (aLRT) bootstrap replicates and approximate Bayes (aBayes) tests (Guindon et al., 2010; Anisimova et al., 2011). The physical properties of the proteins were predicted the SignalP-5.0 (Almagro Armenteros et al., 2019) (signal peptides) and ExPaSy (Gasteiger, 2003) (molecular weight and isoelectric point) servers.

Mass spectrometry data were searched with SearchGUI (Barsnes & Vaudel, 2018) v3.3.17. Parameters were set as follows: tryptic specificity allowing two missed cleavage; precursor M/Z tolerance of 10 ppm and fragment m/z tolerance of 0.5 Da tolerance for MS/MS ions; precursor charge 2–4; carbamidomethylation of C as a fixed modification and oxidation of M as a variable modification. The sequence databases contained the protein sequences predicted with TransDecoder (Haas et al., 2013) v2.1.0 in the transcriptome assembly of the corresponding species (GEPS01 and GEPV01; (Naumenko et al., 2017), as well as common contaminants from the cRAP database (https://www.thegpm.org/crap/). The SearchGUI output was analyzed and visualized in PeptideShaker (Vaudel et al., 2015) v1.16.44. Peptide-spectrum matches, peptides and proteins were validated at a 1.0% false discovery rate estimated using the decoy hit distribution (decoy sequences were added by PeptideShaker). Only proteins having at least two unique peptides were considered as positively identified. Relative quantities of the proteins were estimated with NSAF. The top protein hits were annotated with the NCBI blast web interfact.

The commands used for data analysis are listed in Text S1 for reproducibility and are also available from GitHub (https://git.io/JfJMP).

Results

Field frequency and quantitative analysis of color morphs in E. cyaneus

To determine the proportion of differently colored individuals within the E. cyaneus population, we sampled 3499 animals within 5 days and visually classified them as orange or blue. The final number of orange animals was 104, translating into a frequency of approximately 3%. These animals and approximately the same number of blue (control) animals were kept in the same water tank to normalize for environmental conditions.

The animals were photographed to obtain a quantitative assessment of their color. We found individuals with different colors from completely blue to completely orange (Fig. 2A), and the RGB color values of the pereon suggested that the colors formed a continuous distribution rather than distinct clusters (Fig. 2B). The pereon R/B color index provided the largest median difference (over two-fold) between the morphs classified by eye (Fig. 2C). The color of the antennae did not differ between the orange and blue individuals (Fig. 2D). At the same time, the antennae color index correlated with the visible heterogeneity of the blue-colored individuals: while the pale blue animals had the lowest antennae R/B color index, the greenish animals of the intermediate submorph had the highest index. As human-based classification turned out to be a reliable proxy, we further compared typical blue- or orange-colored individuals.

Total carotenoid content does not define the body color

At the next step, we checked if the varying color intensity was determined by carotenoid content. Our previous observations showed that the color of animals upon ethanol fixation, which leads to protein denaturation, became the same (example in Fig. 3A), and the visible color intensity of ethanol extracts was also similar between the color morphs. To check this result quantitatively, we estimated the carotenoid content of six individuals from each end of the color distribution (as quantified using R/B pereon values; Fig. 3B). No significant difference in the overall carotenoid content of the body was found (Fig. 3C), indicating that other mechanisms were involved.

Figure 2 Quantitative analysis of E. cyaneus color morphs.

(A) Representative photographs of different morphs. (B) Principal component analysis results of pereon color. (C) Pereon color index in 76 individuals classified by eye into two groups. (D) Antennae color index in the same individuals. ***, p < 0.001. Blue dots with cyan or green border mark pale blue individuals without orange tint and the intermediate ones between the blue- and orange-colored, respectively.

Figure 3 Carotenoid content does not define the color morph.

(A) Representative photographs of the same animals before and after ethanol fixation. (B) Color index of the E. cyaneus individuals used for carotenoid extraction. This panel is based on the part of the data used for Fig. 2C. (C) Carotenoid content of the selected E. cyaneus samples. (D) A precopula of E. vittatus featuring a yellow male and a teal female. (E) Carotenoid content of three teal and five yellow animals.

To additionally check our conclusions, we studied another Baikal endemic species belonging to the same genus, E. vittatus, which has a great variety of color morphs. In this case, we only studied the animals belonging to the most frequent yellow and teal morphs (examples shown in Fig. 3D). Again, the total carotenoid content was similar regardless of the color morph (Fig. 3E).

The level of putative carotenoid-binding proteins correlates with the color morph

However, the color of many crustaceans is determined by carotenoprotein complexes, mainly crustacyanins that extend the yellow-red palette of carotenoids by adding blue and violet hues (Maoka, 2011). These proteins have been explored in many decapod species, but there has been no information about the studied species and very scarce information for amphipods in general.

The hemolymph color, as judged by eye, matched the color group of the individual (Figs. 4A–4C, as well as 4L–4N), but hemolymph protein extracts have much fewer protein spots (Bedulina et al., 2016). Thus, we decided to use hemolymph to look for potential differences in protein composition between differently colored animals. First, we checked that the hemolymph color correlated with the body color (Fig. 4D). Then, we extracted proteins from pooled hemolymph of ten blue or orange individuals of E. cyaneus or four teal or yellow individuals of E. vittatus and used the protein extract for 2D-PAGE (Figs. 4E–4F and 4O–4P, respectively; Fig. S3). We observed two major protein spots with molecular weights below 30 kDa and acid isoelectric points that were present in all samples but were much more abundant (from two-fold to ten-fold difference) in samples of blue or teal animals (Figs. 4E–4F and 4O–4P).

Figure 4 The hemolymph of blue E. cyaneus or teal E. vittatus possesses significantly higher amounts of putative crustacyanin analogs than the hemolymph of orange or yellow animals.

(A–K) show the data for E. cyaneus, while (L–R) show the data for E. vittatus. (A and L) Representative photographs of animals of different color morphs. (B, C, M, and N) Representative photographs of the extracted hemolymph; in each case, the photograph is placed below the photograph of the animal of the same color morph. (D) Correlation between pereon and hemolymph color (quantified as the R/B value) of 15 E. cyaneus individuals (E, F, O, and P) Representative 2D-PAGE showing the difference in optical density of putative crustacyanin analogs. The hemolymph of ten individuals of E. cyaneus or four individuals of E. vittatus was pooled in each gel. Percent values correspond to the relative abundance of the protein spot. (G, H, and I) A representative 1D-PAGE and densitometry results for the 15-kDa and 25-kDa putative crustacyanin analog bands, respectively, for 12 replicates for the bands in E. cyaneus. Uncropped gels are available in Fig. S3. **, p < 0.01. (J, K, Q, and R) Tandem mass spectrometry (MS/MS) peptide coverage of the 25-kDa and 15-kDa putative crustacyanin analogs for E. cyaneus and E. vittatus.

The absence of any other major spots of similar weight allowed us to use 1D-PAGE, which requires less material, for quantification of these two proteins. Analysis of 12 E. cyaneus individuals with 1D-PAGE showed that both bands were less abundant in the protein extracts from the hemolymph of orange individuals that in those from the blue ones (Figs. 4G–4I; Fig. S3) with a three-fold change in median values and p <0.01. Interestingly, the relative abundances of the approximately 15-kDa and the 25-kDa bands were strongly correlated (Fig. S4A).

The putative carotenoid-binding protein bands were excised from gels and identified using LC-MS/MS. Top protein hits (>50 unique peptides matched; Table S3) were indeed proteins with predicted low molecular weights, acid isoelectric points and signal peptides (Figs. 4J–4K and 4Q–4R). To our surprise, the best blast hits for obtained sequences were unknown proteins from other crustacean species (Table S4), and no similar proteins from the Uniprot database were found. The only annotated protein hit with e-value <10−20 was a predicted pheromone/general odorant-binding protein from an amphipod Trinorchestia longiramus (KAF2366110.1; Patra et al., 2020). It is important to note that different proteins of the pheromone/odorant-binding family bind a wide range of hydrophobic molecules and function beyond chemoreception (Pelosi et al., 2018).

To check if the 15-kDa and 25-kDa proteins, the amount of which correlated with the color morph, indeed bound to carotenoids, we used native electrophoresis of hemolymph extracts. Indeed, we found some blue and orange protein bands, with blue bands being the most prominent in blue-colored E. cyaneus individuals (Fig. S5). Moreover, blue bands excised from the first direction gels produced 25-kDa and 15-kDa bands among others (Fig. S5), and the only proteins of such weights in hemolymph were those analyzed with mass spectrometry. The presence of higher molecular weight proteins in blue bands may hint at the presence of higher molecular weight complexes of these proteins or other proteins also binding to carotenoids.

These data suggest that the color of an individual correlates with the amount of some carotenoid-binding proteins (and is probably determined by it), even though the particular mechanism requires further investigation. So, the results of the biochemical analyses were overall similar for the two species, as the color morph strongly correlated with the amount of putative carotenoid-binding proteins. As these proteins are not very similar to classical crustacyanins, we suggest calling them crustacyanin analogs.

Phylogenetic analysis suggests that the diversity of coloration- forming proteins in Amphipoda formed independently from that in best-studied Decapoda

Finally, we placed the identified putative crustacyanin analogs within the broader picture of known and putative invertebrate crustacyanins (Fig. 5). The known sequences included crustacyanins A and C from penaeid shrimps (Fenneropenaeus merguiensis, Litopenaeus vannamei and Penaeus monodon) (Budd et al., 2017) and the lobster H. gammarus (Keen et al., 1991a; Keen et al., 1991b). To compare our results with the published phylogeny of crustacyanins (Wade et al., 2009), we included the same sequences from G. pulex, as well as putative carotenoid-binding proteins from blue copepods (Acartia fossae) and appendicularia (Oikopleura dioica) (Mojib et al., 2014), and also searched the GenBank database for other sequences annotated as crustacyanins. Among them, we found sequences from a brachyuran crab Eriocheir sinensis and five sequences annotated as crustacyanins in the genome of a talitrid amphipod Hyalella azteca (Poynton et al., 2018).

Figure 5 The diversity of predicted crustacyanin-like sequences in some amphipod species in comparison with those from different groups of decapods, as well as some copepods and appendicularia.

(A) A maximum-likelihood tree of predicted protein sequences. The shapes are added to visualize the taxa; the blue shapes mark proteins that are potentially coloration-related, while the gray shape marks an apolipoprotein D sequence not related to coloration. aLRT, Shimodaira-Hasegawa approximate likelihood ratio test; aBayes, approximate Bayes test. (B) Abundance of the transcripts encoding the putative crustacyanin analogs and crustacyanin-like proteins in E. cyaneus transcriptome samples. Each dot is one sample (pooled material of 5 animals). TPM, transcripts per million.

In addition, we wanted to know if the studied species possess proteins more similar to decapod crustacyanins than the ones identified with mass spectrometry. To predict putative crustacyanin sequences, we looked for sequences similar to H. gammarus crustacyanins A and C in the published E. cyaneus assembly (GHHW01; Drozdova et al., 2019) and E. vittatus assembly (GEPV01; Naumenko et al., 2017). Then the protein sequences were predicted and re-classified against the non-redundant NCBI protein database. Six distinct sequences that had crustacyanins in the top ten hits were found only in the former. We supposed that, as the assemblies were filtered to remove contamination (Naumenko et al., 2017), some sequences may have failed to pass this filtering or expression level filtering, and thus we reassembled the transcriptome. In the new assembly, we found nine transcripts encoding three distinct putative crustacyanin-like proteins of E. vittatus.

The phylogenetic analysis (Fig. 5A) revealed that all of the sequences from Er. sinensis or amphipod species formed an outgroup for the A and C subunits in penaeid shrimps and lobsters. The same was true for the sequences from copepods and appendicularia, corroborating the original result (Mojib et al., 2014). The sequences revealed with the mass spectrometry analysis clearly form an outgroup to all other sequences. We suggest that the putative crustacyanin analogs also bind carotenoids producing blue-colored complexes, but formed independently of crustacyanins from another family of proteins binding hydrophobic molecules, odorant-binding proteins. Overall, these data may mean that proteins forming blue complexes with carotenoids emerged at least three times even within Crustacea and originated from at least two different protein families, lipocalins and odorant-binding proteins.

However, if the putative crustacyanin analogs, belonging to the odorant-binding family, are major proteins determining the color of the hemolymph, the function of crustacyanin-like sequences from these species remains even more elusive. It is worth noting that all the crustacyanin-like sequences had acid isoelectric points and predicted molecular weights below 30 kDa, corresponding quite well to the observed protein spots. Thus, we hypothesized that the spots could be mixtures of both classes of proteins. However, the crustacyanin-like proteins were not present, even in minor amounts, in mass spectrometry data. Another possibility could be that the crustacyanin analog proteins were restricted to hemolymph. To check for that, we compared the expression levels in 25 transcriptomic samples of E. cyaneus in control conditions published earlier (Drozdova et al., 2019). We found that the crustacyanin analogs had much higher expression levels that were at least two or three orders of magnitude higher than those of the crustacyanin-like transcripts (Fig. 5B). We suggest that the crustacyanin analogs play the main role in determining the body color, while the crustacyanin-like proteins play another role, which is still to be revealed.

Discussion

In this work, we studied the molecular basis of color formation in two species of endemic Lake Baikal amphipods, E. cyaneus and E. vittatus.

In these species, the carotenoid content was not the driving force of color distinction, as it was very similar between animals of contrasting color morphs (Fig. 3C and 3E). This distinguishes the mechanism of intraspecies color variability in Baikal endemic species from that in some other known examples. For example, blue individuals of G. lacustris, another gammaridean amphipod, differ from the usual greyish-brown ones by acanthocephalan infection and reduced carotenoid content caused by the infection (Hindsbo, 1972). Another example of color morphs differing by the level of carotenoids is penaeid shrimp Fenneropenaeus merguiensis (Ertl et al., 2013), but in this case, both crustacyanin and carotenoid levels, as well as other factors, contributed to the formation of three morphs. Instead, we found that the presence of contrasting color morphs is most probably linked to carotenoid-binding proteins (Fig. 4) but not to the total carotenoid content (Fig. 3).

For one of the species, E. cyaneus, we estimated the frequency of differently colored animals. Orange-colored individuals of E. cyaneus were relatively rare (∼3%) in our sample. Still, there are important questions of how much this distribution may vary depending on the sampling place and environmental conditions and how much the color may change throughout the life span of an individual. We did not observe any noticeable changes in color while keeping animals for several months fed ad libitum, as well as any striking difference in size or sex distribution of blue and orange animals, but a deeper analysis is needed to draw informed conclusions. As differently colored individuals coexist in the same microhabitats and most probably have access to the same resources, the mechanism of this difference should have genetic control. It was earlier suggested (Timoshkin, 2001) that color morphs in E. cyaneus exist as a two-allele system with heterozygotes being the fittest. The genetic control of this trait might be even more complex to form the observed continuous variability in E. cyaneus (Fig. 2) and the greater variability of color morphs in E. vittatus (Fig. 1), and constitutes another interesting direction of further research. These data also raise the question about the mechanism of color formation in another Eulimnogammarus species with intraspecies color polymorphism, E. messerschmidtii, which is quite similar to E. cyaneus and has a similar blue/orange color polymorphism, but in the former species, the orange morph is more common (Bedulina et al., 2014).

While ecological aspects may contribute the frequency of differently colored individuals in E. cyaneus, the observed intraspecific color variability correlates with the abundance of particular protein spots (Figs. 4E–4I), and, importantly, these proteins migrated in the colored bands in native electrophoresis (Fig. S5). The same tendency was observed for E. vittatus (Figs. 4O–4P). Identification of these proteins with mass spectrometry revealed that they did not belong to the lipocalin family, similar to known crustacyanins; instead, they were similar to some amphipod proteins annotated as belonging to the pheromone/odorant-binding proteins. These proteins have a structure with a hydrophobic cavity and can bind a wide range of hydrophobic molecules (Pelosi et al., 2018), so the idea that they might bind carotenoids is plausible. We hypothesize that these proteins play the role of amphipod crustacyanins and suggest calling the proteins of this group crustacyanin analogs.

This finding leads to the question of whether amphipods possess homologs of decapod crustacyanins. To answer this question, we searched the transcriptomes of the studied species for sequences similar to lobster crustacyanins. These crustacyanin-like sequences, as well as five sequences annotated as crustacyanins in the genome of H. azteca, an amphipod species belonging to another suborder, and two expressed sequences from a more closely related G. pulex formed a sister group to crustacyanins A and C. Interestingly, the crustacyanin-like sequences from a brachyuran crab Er. sinensis also did not belong to the A or C subunit groups (Fig. 5A). A similar analysis of blue-colored plankton species (a copepod Acartia fossae and an appendicularian Oikopleura dioica) also revealed proteins of the lipocalin family forming an outgroup to the A and C crustacyanin subunits (Mojib et al., 2014). As our analysis, similar to the published ones, included sequences from full transcriptomes and a genome, it is unlikely that some sequences more similar to A- or C-crustacyanins were missed in amphipods. We can safely assume that at least some species belonging to various groups of invertebrates (even decapods) exploit a similar mechanism to the lobster and shrimp but use some other proteins, and the details of their action are a promising direction for future research.

The function of the crustacynin-like proteins in the studied amphipod species is so far unclear. Their predicted physical characteristics (molecular weight and isoelectric point) are very similar to those of crustacyanin analogs, which we studied with mass spectrometry. As no trace of crustacyanin-like proteins was found in the mass spectrometry data, we conclude that they are absent at least from the hemolymph. To check if the crustacyanin analog proteins were specific to the hemolymph, we calculated their expression levels in published transcriptomic samples of E. cyaneus and found that the crustacyanin analogs had two or three orders of magnitude higher expression than the crustacyanin-like sequences (Fig. 5A). Thus, we suppose that the crustacyanin-like proteins have a very specific function confined to particular organs or tissues, while the crustacyanin analogs sufficiently contribute to the visible body color.

The molecular-level mechanism of color formation is still an open question. Lobster proteins act as octamers of heterodimers (Chayen et al., 2003), and the crustacyanins of penaeid shrimps probably act in a similar way, as they form two clear clusters on the phylogenetic tree (Budd et al., 2017). However, there is no available information on subunit composition except for the two distinct carotenoid-binding proteins forming blue complexes in G. lacustris (Czeczuga & Krywuta, 1981). In both studied species, we observed two groups of subunits differing in molecular weight (Figs. 4E–4G, and 4O–4P), and the relative amount of both groups were strongly correlated both on the protein (Fig. S4A) and transcript (Fig. S4B) levels. These data, together with the presence of various bands forming a ladder on the second (denaturing) direction of native 2D-PAGE (Fig. S5), hint at complex formation by these proteins, but this hypothesis requires further investigation.

Conclusions

Here we characterized the coloration of two Baikal amphipod species with intraspecies color morphs. We found that the coloration did not depend on the total carotenoid content, but correlated with the level of putative carotenoid-binding proteins. These proteins, which we suggest terming crustacyanin analogs, are not orthologous to the A and C crustacyanin subunits of lobsters and shrimps, but are related to pheromone/odorant-binding proteins. We suggest that crustacyanin analogs act similarly to the well-studied crustacyanins. However, the details of their action, such as binding to particular carotenoids and the composition of complexes they may form, are still to be revealed.

Supplemental Information

Supplemental Information 1 Precopulae of E. cyaneus

All animals originate from the same catch.

Click here for additional data file.

Supplemental Information 2 E. cyaneus color quantification

(A) Example of a photograph used for color quantification. (B) Gut color index. (C) Pereopod color index. See Fig. 2 for color codes.

Click here for additional data file.

Supplemental Information 3 Uncropped gels for Figs. 4C, 4E, and 4I

Click here for additional data file.

Supplemental Information 4 Relative abundances of the approximately 15-kDa and 25-kDa bands in one-dimensional gels (A) and the corresponding transcripts in the transcriptomes (B)

Click here for additional data file.

Supplemental Information 5 Native PAGE result

The PageRuler Prestained Protein Ladder, 10 to 180 kDa (Thermo Scientific, USA) was used in both native and denaturing gels, but its mobility in native gels is unknown.

Click here for additional data file.

Supplemental Information 6 The correspondence between E. cyaneus samples used for different analyses

Click here for additional data file.

Supplemental Information 7 Densitometry analysis of 1D-PAGE and 2D-PAGE results

Click here for additional data file.

Supplemental Information 8 PeptideShaker summary of peptide and protein matches

Click here for additional data file.

Supplemental Information 9 NCBI blastp summary

The protein sequences were used as a query against the NCBI non-redundant protein database. The analysis was run through the NCBI blast web interface on April 03, 2020.

Click here for additional data file.

Supplemental Information 10 The code used for data analysis

Click here for additional data file.

Supplemental Information 11 Sequences of putative crustacyanins found in the E. cyaneus transcriptome

Click here for additional data file.

Supplemental Information 12 Sequences of putative crustacyanin analogs and crustacyanin-like proteins found in the E. vittatus transcriptome

Click here for additional data file.

Supplemental Information 13 The alignment used to create the phylogenetic tree (Fig. 5A)

Click here for additional data file.

Mass spectrometric analysis was carried out using the equipment of the “Human Proteome” Core Facility of the Institute of Biomedical Chemistry (Moscow, Russia). We are grateful to the staff of the Facility and personally to Dr. Olga Tikhonova for their help. We would like to thank the members of the “Biosystems’ adaptation” lab, as well as to Polina Lipaeva and Timofey Prodanov, for their help in sampling. We are grateful to Dr. Ekaterina Govorukhiva for her help in species identification. Special thanks to Ekaterina Madyarova for laboratory assistance, Ekaterina Shchapova for help with illustrations, Kseniya Vereshchagina for sharing photos of amphipods, and Anton Gurkov for fruitful discussions. Last but not least, we would like to thank the reviewers, Dr. Thomas Knigge and Dr. Nick Wade, for their help in improving the manuscript.

Additional Information and Declarations

Competing Interests

Author Contributions

Data Availability

The authors declare there are no competing interests.

Polina Drozdova conceived and designed the experiments, performed the experiments, analyzed the data, prepared figures and/or tables, authored or reviewed drafts of the paper, and approved the final draft.

Alexandra Saranchina performed the experiments, analyzed the data, authored or reviewed drafts of the paper, and approved the final draft.

Mariya Morgunova, Yulia Lubyaga and Boris Baduev performed the experiments, authored or reviewed drafts of the paper, and approved the final draft.

Alena Kizenko analyzed the data, authored or reviewed drafts of the paper, and approved the final draft.

Maxim Timofeyev conceived and designed the experiments, analyzed the data, authored or reviewed drafts of the paper, and approved the final draft.

The following information was supplied regarding data availability:

The commands used for data analysis are available in GitHub (https://git.io/JfJMP) and the Supplemental File. The raw densitometry measurements and the raw gel images are available in the Supplemental Files.

The mass spectrometry proteomics data are available in the ProteomeXchange Consortium at PRIDE: PXD018516.

http://proteomecentral.proteomexchange.org/cgi/GetDataset?ID=PXD018516.

https://www.ebi.ac.uk/pride/archive/projects/PXD018516.

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
