# Peer review of "The level of putative carotenoid-binding proteins determines the body color in two species of endemic Lake Baikal amphipods"

_PeerJ, doi:10.7717/peerj.9387_

## Round 0.1 · original submission · Major Revisions

Dear Authors

The reviewers have recommended considerable and major revisions to your manuscript and have provided two extensive reviews which I hope you find will improve the clarity of the manuscript. I would be happy to receive a revised version if you feel you can address the issues raised.

d·

Basic reporting

The article structure follows conventional standards of scientific publication (good written English, figure quality, article structure etc.), but the authors include elements that should be provided in the Material and Methods section within the Results section, which is confusing.

Experimental design

The research question is not well defined and this evident for the introduction and the discussion section. The authors are not clear about what they are studying, which is the molecular formation of colour, but not the polymorphism. Their study objective to “uncover the mechanism underlying the formation of color polymorphism in E. cyaneus and E. vittatus” is incorrect, because they have investigated the molecular mechanisms of color formation (correctly stated at the beginning of the discussion). Consequently they authors are also drawing false or unsupported conclusions about the role of crustacyanins in colour polymorphism. The research gap is stated (lack of knowledge concerning colour formation in amphipods), but this is partly gone to water with the considerations about polymorphism. For parts of the study, the technical standards are perhaps suboptimal and need confirmation by additional work.

Validity of the findings

The novelty of the findings is limited as the mechanisms of colour formations are basically known and have been more or less confirmed here for amphipods. Some of the conclusions are, however, not valid as much as polymorphism is concerned. The authors have been comparing blue and orange morphs of two gammarid species. I do not think that this can be considered studying the mechanisms underlying polymorphism in these species and the focus should, therefore, rather be on the molecular role of crustacyanin in the formation of different colours in amphipods as compared to other crustaceans. With their focus on polymorphism the authors also tend to draw false and/or unsupported conclusions about the role of crustacyanins in producing highly polymorphic amphipods (for instance there are no data whether different populations of the studied amphipods would vary in polymorphism, as could be established by a Shannon-Wiener index, and how this relates to variability in the expression and concentrations of crustacyanins in the various morphs).

Additional comments

The manuscript PeerJ 44934-v0 describes the contribution of presumed amphipod crustacyanins to the formation of different colour morphs in two grammarids of Lake Baikal. The authors investigate the molecular mechanisms of colour formation in two species of the genus Eulimnogammarus from Lake Baikal, which has a particularly rich and polymorph diversity of amphipod species. The manuscript is well written, but the structure lacks clarity, particularly because the Material and Methods section lacks information that is contained in the results part. Furthermore, the objectives of the study are not clearly defined. Especially, the expression “mechanisms forming … diversity” or “mechanism underlying the formation of colour polymorphism” are, to my point of view, misleading. The scope of the study is much more limited and describes a putative mechanism of how different colours are formed in these animals, which, of course is a prerequisite for the observed colour polymorphism, but it is not the mechanism responsible for the polymorphism. Knowing how the colours are formed does neither explain within population variability in colour morphs, nor why these amphipods produce so many different colour morphs. In other words, it does not answer the question why the many different amphipod species in Lake Baikal are highly polymorphic. The mechanisms for this remarkable inter- and intraspecies polymorphism are not the capacity of forming different colours, but rather the interactions of these amphipods within a population and with their specific environment: location of habitat, depth, temperature, exposure to different predators, i.e., ecological and population parameters that trigger colour polymorphism.

Considering the article structure, the introduction describes this specific polymorphic composition of the vast amphipod fauna in Lake Baikal, which provides the background for the study. This done, however, too extensively and, more importantly, it should be clearly distinguished from the question of how the colours are formed. Within the introduction, the authors start with some information about the elements of colour formation (carotenoids, i.e., astaxanthin, crustacyanin) and their supposed roles in colour formation, but this aspect, which is fundamental to study, needs to be treated in more depth (e.g. see Cianci et al., 2002, Wade et al. 2005). The authors should inform how and where colours are formed in crustaceans. They should also point to the fact that they are dealing with morphological shell coloration. With regards to the method section, it would be interesting to know, what the authors are actually measuring (carotenoids from the shell, whole body, haemolymph) and how the haemolymph concentrations relate to the concentrations in the cuticle, which eventually determine crustacean shell colour. The authors, however, switch to the polymorphisms in Lake Baikal amphipods. What is further lacking in this context is a link how the authors consider the molecular mechanisms of colour formation to be linked to the polymorphisms. Hence I would recommend beginning the introduction with some remarks on the astonishing variability of coloration amongst Lake Baikal amphipods and the intraspecies polymorphism making it worthwhile to study the phenomena of colouration in these amphipods. The authors should explain to the reader what is the nature of this coloration (morphological pigmentation of the cuticle incorporating pigments from the epithelium) and how astaxanthines and crustacyanins interact with each other to produce different colours. Then they could continue with their statement, that a lack of knowledge exists with regards to these pigment and protein molecules in amphipods, providing the justification for the study.

The Material and methods section also lacks clarity. Unfortunately, a good amount of relevant information has been included into the results section. For instance, l. 196-198 explains the amount of animals samples, how they were classified and sorted, which is information that should be provided in ‘Animals and sampling’, as should be the information provided in l. 203-205. L. 206 talks of an experimental setup, but from the Material and methods section it is not clear that an experiment was carried out and what would be the nature of it. That animals were photographed after acclimation period also is methodological information. The paragraph ‘Carotenoid measurements’ mentions ‘samples’, but the authors do not specify what type of samples. From the results part it becomes clear that these samples are whole animals that were fixed in either ethanol or liquid N2. As the final colour is formed in the cuticle, why do the authors not study the carotenoids in the cuticle (extraction from shell as in Wade et al. 2005)? Similarly, why are crustacyanins from the haemolymph quantified although these are, most likely not involved in colour formation in the cuticle and why do the authors not extract crustacyanins from shell (e.g. Cianci et al., 2002). Can the authors ensure that both levels of crustacyanin are at least correlated? From the paragraph on electrophoretic methods it remains unclear what the 1D- and the 2D-PAGE are used for? It is quite surprising to learn from the results section that 2D gel electrophoresis is only used for a kind of pseudo-identification. Even if I follow the argumentation of the authors about the MW and pI matching theoretical values, it is difficult to accept that the authors have not confirmed the identity of the respective spots by MS/MS. The same holds true for the quantification of the 25kDa and 15kDa bands. Please understand that I do not question the assumptions made. The argumentation made by the authors is plausible (l. 249-250). Nevertheless, you are more or less blindly quantifying bands the nature of which has been inferred from secondary information. Furthermore, the manuscript would have gained impact if the authors could confirm the AA sequences and compare them with those retrieved in genbank. It should not be too difficult to find a group of mass spectrometrists that would analyse the respective spots. To my liking the proposed article presents too much ‘predicted’, suggested and ‘putative’ findings, which significantly lessens its overall impact. Given the coherence of the findings, I do not doubt the general conclusions of the study, but the relatively simple mass spectrometric analysis of four spots plus four bands would greatly increase the novelty, and thus the value of this study! Furthermore, the authors should demonstrate that their quantitative analyses from heamolymph do indeed reflect the levels of crustacyanins in the shell/cuticle, to ensure the validity of their measurements. Again the results are coherent and the conclusions drawn are plausible, but not verifying the correlation between the concentrations of crustacyanins in the cuticle and heamolymph and how these are related it is at least contestable. According to Ertl et al. (2013) the major site of crustacyanin expression is the endocuticle/outer epithelium and hence they are likely to be synthesised therein, or according to other sources in the subepidermal adipose tissue. But Ertl et al (2013) also found crustacyanin to be expressed in other tissues. It is therefore not straightforward to measure haemolymph crustacyanins as a surrogate of cuticle crustacyanis. It is also unclear how various physiological parameters or moult status would influence the concentrations of circulating crustacyanin. Hence, numerous open-ended questions may cast doubt over this method.

Eventually, the discussion is not strictly related to the role of astaxanthine and crustacyanin in the formation of shell colour. Here the manuscript suffers from the same lack of clarity already stated for the introduction and enters into a discussion about, how colour may change over lifetime, ecological aspects that contribute to polymorphism in Eulimnogammarus and relates the presence of crustcyanins to this polymorphism. To my point of view the authors mix the mechanism of forming blue colour in crustaceans, with mechanisms that produce the observed polymorphism. This is by no means the same! Furthermore, the authors claim that not the carotenoids are responsible for the colouration, but they are not precise about their comparison: the amount of asthaxanthin in the cuticle is certainly responsible for the degree of coloration between paler or more intensively coloured (orange) morphs (e.g. Wade et al. 2005). When it comes to blue coloured morphs, this requires molecular modifications of the astaxanthins to obtain a different absorption spectrum and in this case the crustacyanins are crucial for producing blue colour. Hence the difference between blue and orange morphs does not stem from the amount of carotenoids and this would indeed not be expected. It is known that he crustacyanins change the colour of astaxanthins and the results of the present study confirms this principle for the studied amphipods, thus increasing evidence that this is a general mechanism, even if some difference in the crustacyanin sequences exist. The authors should rather discuss these molecular mechanisms, which they have been studying. A good example is given by the penaeid shrimp F. merguiensis (Ertel et al. 2013). As the authors mention in their discussion the formation of colour morphs is a complex interplay between astaxanthin, crustacyanin and further factors. Hence, their conclusion that the carotenoid content is unimportant for the diversity of colour morphs (l. 312) cannot be generalised from the results of this study.

Furthermore, the capacity of forming blue, orange and other colours is not the reason for the observed polymorphism. The crustacyanins are simply a necessity to form complexes with the asthaxanthins that bend them into different orientations so as to change their absorption and produce different colours. But why these colours are produced in some morphs or in some body parts and not in others has not been studied and the discussion of these aspects is, therefore, not supported by any data of this study. The authors do not produce results that could explain the reasons for the polymorphism of these amphipods, i.e., why crustacyanins are produced at a higher degree in some morphs and to a lesser degree in others. Therefore, the discussion about polymorphism is distracting from the essence of the study, which was to confirm the role of crustcyanins in the formation of blue coloured amphipods. As the mechanism of blue colour formation has been described earlier (e.g. Krawczyk and Britton 2001, Chayen et al. 2003) and the extracted putative crustacyanins have been confirmed only indirectly the novelty of the proposed publication is, unfortunately, greatly reduced.

In summary, it is strongly recommended to repeat the gels with sufficient material for subsequent AA sequence analyses by MS/MS and to focus the introduction as well as the discussion on the molecular mechanisms of colour formation in crustaceans across different taxonomic groups (decapods, amphipods, …). If the MS/MS analyses cannot be carried out, the authors should at least whet the objectives of the study within the limits of what has been studied and avoid deviating from the molecular formation of different colours in the two gammarids into general considerations of polymorphism, which are not supported by the data. Nevertheless, the authors may, of course, highlight that the many highly polymorphic species of Lake Baikal provide a particularly interesting study subject.

·

Basic reporting

The present study assessed the underlying molecular mechanisms that explain colour morphs in freshwater amphipods. The images are excellent and helpful, but the organisation of the Figures and the way in which the Results are presented is not so logical, for me at least.

1. This entire Results section needs to be written more directly and in past tense. For example: L208-210 “We found that the individuals of this species had different colors, from completely blue to completely orange (Fig. 2A), and the metrics of R, G, and B colors of the pereon corroborated what was visible by eye (Fig. 2B).” and L221-225 “Six individuals from each end of the color distribution (as quantified using R/B values, Fig 3B) were used to quantify their carotenoid content. No significant difference in the overall carotenoid content of the body was found between these colour morphs (Fig. 3C), and thus indicated other mechanisms were involved.” There are many more examples throughout other sections, partly caused by the arrangement of data (see next comment).
2. Once arriving at Fig 4 and 5, the Results begin to combine sequence comparisons and protein blots from one species, then repeats several colour analyses, carotenoid and protein extractions from another. In my opinion, it would be better to put the animal descriptions (for both species) up front in Figure 2, then organise the proceeding Figures more consistently to contain shared data from both species in successive Figures. This might be something like: PCA of RGB values in one Figure, carotenoid quantification in another, then protein blots. I question the need for phylogenetic analysis as part of these Figures, as this is covered in Figure 6. As presented, the organisation of the Figures does not allow for a very easy description of Results and leads to confusion in the Results section, plus considerable Discussion points in the Results. This suggested re-organisation is supported by the authors own statement on L277-279, that Results are similar for both species. Perhaps consider a combined Results and Discussion section?

Experimental design

An impressive series of experiments was conducted on a species with a distinct colour morph, with the clear goal to understand the role or protein bound carotenoids to produce their colouration. In addition, further investigation of CRCN sequences from a range of crustaceans was performed, and further resolved the diversity and evolutionary significance of this protein family. The authors should be commended for including their coding as part of the manuscript, this is often overlooked. The extraction of hemolymph from such small animals also shows great technical skill.

3. Please provide a more detailed description of the method used to quantify colour from digital images.
4. The methods are well explained but would benefit from clearly stating how many individuals were used for each method. This would clearly describe how many individuals were used for each of the colour quantification, carotenoid extraction or protein gel sections. If the same individuals were used in different sections this should also be made clear, as this can strengthen the associations between the data sections.
5. Please describe the use of coloured dots in Figure 2 as part of the Figure legend. However, only two pale blue and intermediate blue/orange individuals were shown on Figure 2. I question the need, ability or importance to isolate them from the blue and orange forms. Were these individuals only defined by eye, or was there supporting RGB values that could classify them? Without support for the additional colours based on RGB quantification, it might be best to remove the two individuals from the other graphs and just use the individuals from the blue and orange dots.
6. The field frequency data is useful, but less so the survival information provided as part of Supplementary File S3. If not replicated and significantly different between the two morphs (which was not stated or evident from the data presented), this would be the weakest part of the study and might be best left out so remove lines 203-206.
7. Although quite distinct species, the past study by Mojib et al 2014 (Molecular Ecology 23: 2740-2756) should be included in the phylogenetic analyses and Discussion, and mentioned in the Introduction as well. The present work builds on this nicely to demonstrate the expansion of CRCN genes among animals.
8. It was not clear whether the protein alignments used full length sequences, which could be made clearer either by denoting the corresponding residues from the reference sequence (as part of the Methods) or include the alignment file used for the phylogeny as a Supplementary file.

Validity of the findings

It is very interesting to me the appearance of several CRCN genes in diverse crustacean lineages, broadening the functional diversity of this family. The conclusion that these proteins are acting in the same way to produce colour despite considerable sequence diversity is solid.

9. L43-48 – several studies in decapod crustaceans have shown that pigment levels are not correlated with body colour, particularly in response to environmental cues. The study by Ertl was the exception to others such as Tume et al 2009 (Aquaculture 296: 129-135) and Wade et al 2015 (Aquaculture 449: 78-86). Indeed, the latter study quantified CRCN abundance using an antibody

Additional comments

It would be interesting to know whether the carotenoids extracted from red morphs were esterified, as this was found to be the case for Penaeid prawns that had removed their crustacyanin. Perhaps more a comment for future work than a requirement for modifying this work, this can be done relatively simply using thin layer chromatography or HPLC if you have one.

Would be nice to crystalise these proteins, or maybe even just predict their 3D structure based on decapod CRCN!

Minor comments
• Line 123 One blue and one orange individual was…..
• Line 207 I would say this method was quantitative. You should remove “rough”
• Exact significance values are not important, and it is sufficient to use a P value cutoff of 0.05 or 0.001. Non significant P values do not need to be reported at all
• The arrows in Figure 4B don’t quite align with the protein spots
• It would be helpful to define aLRT in the Figure 6 legend, and explain the associated red dots on the phylogeny.
• Lines 360-364 has some awkward wording, perhaps divide into several sentences.

---

## Round 0.2 · Minor Revisions

Dear Authors

Thank you for your efforts in revising this manuscript. As you can see from the reviewers comments they were impressed by the revision and improvements made. The decision is accept but I have selected "minor revisions" as one reviewer made an additional suggestion and a few typos which you may wish to consider.

·

Basic reporting

The authors write a correct scientific English that describes the research subject, the methods used as well as the results and corresponding discussion in a clear and unambiguous manner

The introduction of the revised manuscript is focused, places the research work in a clearly defined context and delivers the necessary background in a concise, but comprehensive way. The cited literature is pertinent and references the current state of the art.

The structure of the manuscript conforms the discipline norm of scientific publications in the field of life sciences and has been greatly for improved for clarity as compared to the original manuscript.

All figures are relevant and of exceptional quality. The figures are extensively labelled and described correctly.

The data supplied are ample and exhaustive. Extensive supplementary data are provided, but mass spectrometry data seem not to be included.

Experimental design

The manuscript describes original primary research that is within the scope of the journal.

The research question is now defined more clearly. The authors have clearly outlined the lack of knowledge for the taxonomic group subject to their investigations, as compared to the decapods, where coloration has been studied in more detail. Hence, the research question is relevant and meaningful. The results obtained in this study also underscore the research need with regard to the mechanism of colouration in species less well studied.

The investigations performed are scientifically sound and of high technical standard.

The methods are described with sufficient detail and should be reproducible. The material and methods section has been improved as compared to the original manuscript. Given the impressive amount of work accomplished at different levels (sampling, image analyses, protein biochemistry, molecular biology) this section naturally has to be quite voluminous.

Validity of the findings

The data supporting the conclusions have been provided exhaustively; they appear to be coherent and reliable.

The impact and novelty of the data is highlighted in the results and discussion sections. By including the results of mass spectrometric analyses, the manuscript has considerably gained in impact and novelty, because the results revealed an unexpected nature of the proteins. These surprising results open new research questions and should be very interesting to the scientific community working in the field. It is understood, that the new proteins revealed by this study, need to be confirmed and their role remains, for the time being, speculative. The authors have made this perfectly clear.

The conclusions in the revised version of the manuscript are well stated, clearly linked to original research question described in the introduction and based on the outcome of the study.

Additional comments

First of all I would like to express my respect for the impressive amount of work the authors have invested into this study and congratulate them for the additional work that they have accomplished in a little while. This is a magnificent example of how to revise a manuscript! . By my reckoning, the investment into mass spectrometry was worth the while and has both increased the novelty and impact that the paper will have.
Indeed, the manuscript has exquisitely evolved and matured to an impactful research work that should make an excellent contribution to the scientific literature on colouration in crustaceans. The authors have responded perfectly to the questions of the reviewers. For my part, I have no further questions or comments. I can only applaud the authors for the work they accomplished and wholeheartedly recommend the manuscript for publication!
However, I would suggest to improve the title and suggest something like “Putative carotenoid-binding proteins determine colour formation in two species of endemic Lake Baikal amphipods”. Furthermore, there is a typo in the abstract (L.41 “then” should be “than”). I would also like to suggest some corrections to the figure legends:
Rather use the term “Representative photographs” in the legend to figure 4, as in the legends to figure 1, 2 and 3 and replace “example” by "representative", i.e., "representative 2D-PAGE/1D-PAGE". Perhaps specify (MS/MS) for the mass spectrometric analysis. In the legend to figure 5, replace “some amphipod species” by “selected amphipod species”.
I will take great pleasure to see this manuscript published and congratulate the authors for their work! The acknowledgements for reviewing this article are flattering, but would have not been necessary. I did my best and if this was helpful the sense of reviewing was fulfilled.

---

## Round 0.3 · accepted · Accept

Dear Authors

Thank you for the revised manuscript.